# Tree-Sliced Variants of Wasserstein Distances

**Tam Le**
RIKEN AIP, Japan
`tam.le@riken.jp`

**Makoto Yamada**
Kyoto University & RIKEN AIP, Japan
`makoto.yamada@riken.jp`

**Kenji Fukumizu**
ISM, Japan & RIKEN AIP, Japan
`fukumizu@ism.ac.jp`

**Marco Cuturi**
Google Brain, Paris & CREST - ENSAE
`cuturi@google.com`

## Abstract

Optimal transport (OT) theory defines a powerful set of tools to compare probability distributions. OT suffers however from a few drawbacks, computational and statistical, which have encouraged the proposal of several regularized variants of OT in the recent literature, one of the most notable being the *sliced* formulation, which exploits the closed-form formula between univariate distributions by projecting high-dimensional measures onto random lines. We consider in this work a more general family of ground metrics, namely *tree metrics*, which also yield fast closed-form computations and negative definite, and of which the sliced-Wasserstein distance is a particular case (the tree is a chain). We propose the tree-sliced Wasserstein distance, computed by averaging the Wasserstein distance between these measures using random tree metrics, built adaptively in either low or high-dimensional spaces. Exploiting the negative definiteness of that distance, we also propose a positive definite kernel, and test it against other baselines on a few benchmark tasks.

## 1 Introduction

Many tasks in machine learning involve the comparison of two probability distributions, or histograms. Several geometries in the statistics and machine learning literature are used for that purpose, such as the Kullback-Leibler divergence, the Fisher information metric, the $\chi^2$ distance, or the Hellinger distance, to name a few. Among them, the optimal transport (OT) geometry, also known as Wasserstein [65], Monge-Kantorovich [34], or Earth Mover's [54], has gained traction in the machine learning community [26, 39, 43], statistics [18, 50], or computer graphics [41, 61].

The naive computation of OT between two discrete measures involves solving a network flow problem whose computation scales typically cubically in the size of the measures [10]. There are two notable lines of work to reduce the time complexity of OT. *(i)* The first direction exploits the fact that simple ground costs can lead to faster computations. For instance, if one uses the binary metric $d(x, z) = \mathbb{1}_{x \neq z}$ between two points $x, z$, the OT distance is equivalent to the total variation distance [64, p.7]. When measures are supported on the real line $\mathbb{R}$ and the cost $c$ is a nonnegative convex function $g$ applied to the difference $z - x$ between two points, namely for $x, z \in \mathbb{R}$, one has $c(x, z) = g(z - x)$, then the OT distance is equal to the integral of $g$ evaluated on the difference between the generalized quantile functions of these two probability distributions [57, §2]. Other simplifications include thresholding the ground cost distance [51] or considering for a ground cost the shortest-path metric on a graph [52, §6]. *(ii)* The second one is to use regularization to approximate solutions of OT problems, notably entropy [14], which results in a problem that can be solved using Sinkhorn iterations. Genevay et al. [26] extended this approach to the semi-discrete and continuous OT problems using stochastic optimization. Different variants of Sinkhorn algorithm have been

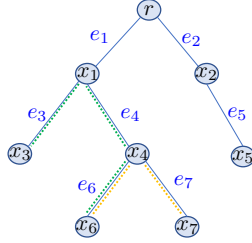

Figure 1: An illustration for a tree with root $r$ where $x_1, x_2$ are at depth level 1, and $x_6, x_7$ are at depth level 3. Path $\mathcal{P}(x_3, x_6)$ contains $e_3, e_4, e_6$ (the green-dot path), $\Gamma(x_4) = \{x_4, x_6, x_7\}$ (the yellow-dot subtree), $v_{e_4} = x_4$, and $u_{e_4} = x_1$.

proposed recently [4, 17], and speed-ups are obtained when the ground cost is the quadratic Euclidean distance [2, 3], or more generally the heat kernel on geometric domains [61]. The convergence of Sinkhorn algorithm has been considered in [4, 25].

In this work, we follow the first direction to provide a fast computation for OT. To do so, we consider tree metrics as ground costs for OT, which results in the so-called tree-Wasserstein (TW) distance [15, 21, 46]. We consider two practical procedures to sample tree metrics based on spatial information for both low-dimensional and high-dimensional spaces of supports. Using these random tree-metrics, we propose tree-sliced-Wasserstein distances, obtained by averaging over several TW distances with various ground tree metrics. The TW distance, as well as its average over several trees, can be shown to be negative definite[1]. As a consequence, we propose a positive definite tree-(sliced-)Wasserstein kernel that generalizes the sliced-Wasserstein kernel [11, 36].

The paper is organized as follows: we give reminders on OT and tree metrics in Section 2, introduce TW distance and its properties in Section 3, describe tree-sliced-Wasserstein variants with practical families of tree metrics, and proposed tree-(sliced)-Wasserstein kernel in Section 4, provide connections of TW with other work in Section 5, and follow with experimental results on many benchmark datasets in word embedding-based document classification and topological data analysis in Section 6, before concluding in Section 7. We have released code for these tools[2].

## 2  Reminders on Optimal Transport and Tree Metrics

In this section, we briefly review definitions of optimal transport (OT) and tree metrics. Let $\Omega$ be a measurable space endowed with a metric $d$. For any $x \in \Omega$, we write $\delta_x$ the Dirac unit mass on $x$.

**Optimal transport.**   Let $\mu$, $\nu$ be two Borel probability distributions on $\Omega$, $\mathcal{R}(\mu, \nu)$ be the set of probability distributions $\pi$ on the product space $\Omega \times \Omega$ such that $\pi(A \times \Omega) = \mu(A)$ and $\pi(\Omega \times B) = \nu(B)$ for all Borel sets $A$, $B$. The 1-Wasserstein distance $W_d$ [64, p.2] between $\mu$, $\nu$ is defined as:

$$W_d(\mu, \nu) = \inf \left\{ \int_{\Omega \times \Omega} d(x, z) \pi(dx, dz) \mid \pi \in \mathcal{R}(\mu, \nu) \right\}. \tag{1}$$

Let $\mathcal{F}_d$ be the set of Lipschitz functions w.r.t. $d$, i.e. functions $f : \Omega \to \mathbb{R}$ such that $|f(x) - f(z)| \leq d(x, z), \forall x, z \in \Omega$. The dual of (1) simplifies to the following problem OT [64, Theorem 1.3, p.19] is:

$$W_d(\mu, \nu) = \sup \left\{ \int_{\Omega} f(x) \mu(dx) - \int_{\Omega} f(z) \nu(dz) \mid f \in \mathcal{F}_d \right\}. \tag{2}$$

**Tree metrics.**   A metric $d : \Omega \times \Omega \to \mathbf{R}$ is called a *tree metric* on $\Omega$ if there exists a tree $\mathcal{T}$ with non-negative edge lengths such that all elements of $\Omega$ are contained in its nodes and such that for every $x, z \in \Omega$, one has that $d(x, z)$ equals to the length of the (unique) path between $x$ and $z$ [58, §7, p.145–182]. We write the tree metric corresponding to that tree $d_{\mathcal{T}}$.

## 3   Tree-Wasserstein Distances: Optimal Transport with Tree Metrics

Lozupone and co-authors [44, 45] first noticed, when proposing the `UniFrac` method in the metage-nomics community, that the Wasserstein distance between two measures supported on the nodes of the same tree admits a closed form when the ground metric between the supports of the two measures is a tree metric. That method was used to compare microbial communities by measuring the phylogenetic distance between sets of taxa in a phylogenetic tree as the fraction of the branch length of the tree that leads to descendants from either one environment or the other, but not both [44]. In this section, we follow [15, 21, 46] to leverage the geometric structure of tree metrics, and recall their main result.

Let $\mathcal{T}$ be a tree rooted at $r$ with non-negative edge lengths, and $d_{\mathcal{T}}$ be the tree metric on $\mathcal{T}$. For nodes $x, z \in \mathcal{T}$, let $\mathcal{P}(x, z)$ be the (unique) path between $x$ and $z$ in $\mathcal{T}$, $\lambda$ is the unique Borel measure (i.e. length measure) on $\mathcal{T}$ such that $d_{\mathcal{T}}(x, z) = \lambda(\mathcal{P}(x, z))$. We also write $\Gamma(x)$ for a set of nodes in the subtree of $\mathcal{T}$ rooted at $x$, defined as $\Gamma(x) = \{z \in \mathcal{T} \mid x \in \mathcal{P}(r, z)\}$. For each edge $e$ in $\mathcal{T}$, let $v_e$ be the deeper level node of edge $e$ (farther to the root), $u_e$ is the other node, and $w_e = d_{\mathcal{T}}(u_e, v_e)$ is the non-negative length of that edge, illustrated in Figure 1. Then, TW not only has a closed form, but is negative definite.

**Proposition 1.** *Given two measures $\mu$, $\nu$ supported on $\mathcal{T}$, and setting the ground metric to be $d_{\mathcal{T}}$, then*

$$W_{d_{\mathcal{T}}}(\mu, \nu) = \sum_{e \in \mathcal{T}} w_e \left| \mu(\Gamma(v_e)) - \nu(\Gamma(v_e)) \right|. \tag{3}$$

*Proof.* Following [21], for any $f \in \mathcal{F}_{d_{\mathcal{T}}}$ such that $f(r) = 0$, there is an $\lambda$-a.e. unique Borel function $\mathtt{f} : \mathcal{T} \to [-1, 1]$ such that $f(x) = \int_{\mathcal{P}(r,x)} \mathtt{f}(z)\lambda(dz) = \int_{\mathcal{T}} \mathbb{1}_{z \in \mathcal{P}(r,x)}\mathtt{f}(z)\lambda(dz)$. Intuitively, $f(x)$ models a flow along the (unique) path of the root $r$ and node $x$ where $\mathtt{f}(z)$ controls a probability amount, received or provided by $f(x)$ on $dz$. Note that $\mathbb{1}_{z \in \mathcal{P}(r,x)} = \mathbb{1}_{x \in \Gamma(z)}$, then we have:

$$\int_{\mathcal{T}} f(x)\mu(dx) = \int_{\mathcal{T}} \int_{\mathcal{T}} \mathbb{1}_{z \in \mathcal{P}(r,x)}\mathtt{f}(z)\lambda(dz)\mu(dx) = \int_{\mathcal{T}} \mathtt{f}(z)\lambda(dz)\mu(\Gamma(z)).$$

Then, plugging this identity in Equation (2), we have:

$$W_{d_{\mathcal{T}}}(\mu, \nu) = \sup \left\{ \int_{\mathcal{T}} \left( \mu(\Gamma(z)) - \nu(\Gamma(z)) \right) \mathtt{f}(z)\lambda(dz) \right\} = \int_{\mathcal{T}} \left| \mu(\Gamma(z)) - \nu(\Gamma(z)) \right| \lambda(dz),$$

since the optimal function $f^*$ corresponds to $\mathtt{f}(z) = 1$ if $\mu(\Gamma(z)) \geq \nu(\Gamma(z))$, otherwise $\mathtt{f}(z) = -1$. Moreover, we have $\mu(\Gamma(r)) = \nu(\Gamma(r)) = 1$, and $\lambda(\mathcal{P}(u_e, v_e)) = d_{\mathcal{T}}(u_e, v_e) = w_e$. Therefore,

$$W_{d_{\mathcal{T}}}(\mu, \nu) = \sum_{e \in \mathcal{T}} w_e \left| \mu(\Gamma(v_e)) - \nu(\Gamma(v_e)) \right|,$$

since the total mass flowing through edge $e$ is equal to the total mass in subtree $\Gamma(v_e)$.  ∎

**Proposition 2.** *The tree-Wasserstein distance $W_{d_{\mathcal{T}}}$ is negative definite.*

*Proof.* Let $m$ be the number of edges in tree $\mathcal{T}$. From Equation (3), $\mu(\Gamma(v_e))$ with $e \in \mathcal{T}$ can be considered as a feature map for probability distribution $\mu$ onto $\mathbb{R}_+^m$. Consequently, the TW distance is equivalent to a weighted $\ell_1$ distance between these representations, with non-negative weights $w_e$, between these feature maps. Therefore, the tree-Wasserstein distance is negative definite[3].  ∎

## 4   Tree-Sliced Wasserstein by Sampling Tree Metrics

Much as in sliced-Wasserstein (SW) distances, computing TW distances requires choosing or sampling tree metrics. Unlike SW distances however, the space of possible tree metrics is far too large in practical cases to expect that purely random trees can lead to meaningful results. We consider in this section two adaptive methods to define tree metrics based on spatial information in both low and high-dimensional cases, using partitioning or clustering. We further average the TW distances

corresponding to these ground tree metrics. This has the benefit of reducing quantization effects, or cluster sensitivity problems in which data points may be partitioned or clustered to adjacent but different hypercubes [32] or clusters respectively. We then define the tree-sliced Wasserstein kernel, that is the direct generalization of those considered by [11, 36].

**Definition 1.** *For two measures $\mu, \nu$ supported on a set in which tree metrics $\{d_{\mathcal{T}_i} \mid 1 \le i \le n\}$ can be defined, the tree-sliced-Wasserstein (TSW) distance is defined as:*

$$\text{TSW}(\mu, \nu) = \frac{1}{n} \sum_{i=1}^{n} W_{d_{\mathcal{T}_i}}(\mu, \nu). \tag{4}$$

Note that averaging of negative definite functions is trivially negative definite. Thus, following Definition 1 and Proposition 2, the TSW distance is also negative definite. Positive definite kernels can be therefore derived following [9, Theorem 3.2.2, p.74], and given $t > 0$, $\mu, \nu$ on tree $\mathcal{T}$, we define the following tree-sliced-Wasserstein kernel,

$$k_{\text{TSW}}(\mu, \nu) = \exp(-t\,\text{TSW}(\mu, \nu)). \tag{5}$$

Very much like the Gaussian kernel, one can tune if needed the bandwidth parameter $t$ according to the learning task that is targeted, using e.g. cross validation.

**Adaptive methods to define tree metrics for the space of support data.** We consider sampling mechanisms to select tree metrics to be used in Definition 1.

One possibility is to sample tree metrics following the general idea that these tree metrics should approximate the original distance [7, 8, 13, 22, 30]. This was the original motivation for previous work focusing on approximating the OT distance with the Euclidean ground metric (a.k.a. $W_2$ metric) into $\ell_1$ metric for fast nearest neighbor search [16, 32]. Our goal is rather to sample tree metrics for the space of supports, and use those random tree metrics as ground metrics. Much like 1-dimensional projections do not offer interesting properties from a distortion perspective but remain useful for sliced-Wasserstein (SW) distance, we believe that trees with large distortions can be useful. This follows the recent realization that solving exactly the OT problem leads to overfitting [52, §8.4], and therefore excessive efforts to approximate the ground metric using trees would be self-defeating since it would lead to overfitting within the computation of the Wasserstein metric itself.

• **Partition-based tree metrics.** For low-dimensional spaces of supports, one can construct a partition-based tree metric with a tree structure $\mathcal{T}$ as follows:

---

**Algorithm 1** `Partition_Tree_Metric(s, X, x̃_s, h, H_𝒯)`

---

**Input:** s: a side-$\ell$ hypercube, $X$: a set of $m$ data points of $\mathbb{R}^{\mathsf{d}}$ in s, $\tilde{x}_{\mathsf{s}}$: a parent node, h: a current depth level, and $H_{\mathcal{T}}$: the predefined deepest level of tree $\mathcal{T}$.
1: **if** $m > 0$ **then**
2:   **if** h $> 0$ **then**
3:     Node $\tilde{x}_c \leftarrow$ a point center of s.
4:     Length of edge $(\tilde{x}_{\mathsf{s}}, \tilde{x}_c) \leftarrow$ distance $(\tilde{x}_{\mathsf{s}}, \tilde{x}_c)$.
5:   **else**
6:     Node $\tilde{x}_c \leftarrow \tilde{x}_{\mathsf{s}}$.
7:   **end if**
8:   **if** $m > 1$ and h $< H_{\mathcal{T}}$ **then**
9:     Partition s into $2^{\mathsf{d}}$ side-$(\ell/2)$ hypercubes.
10:     **for each** side-$(\ell/2)$ hypercube $\mathsf{s}_c$ **do**
11:       $\tilde{X}_c \leftarrow$ data points of $X$ in $\mathsf{s}_c$.
12:       `Partition_Tree_Metric(`$\mathsf{s}_c, \tilde{X}_c, \tilde{x}_c, $h$ + 1, H_{\mathcal{T}}$`)`.
13:     **end for**
14:   **end if**
15: **end if**

---

Assume that data points are in a side-$(\beta/2)$ hypercube of $\mathbb{R}^{\mathsf{d}}$. We then randomly expand it into a hypercube $\mathsf{s}_0$ with side at most $\beta$. Inspired by a series of grids in [32], we set the center of $\mathsf{s}_0$ as the

root of $\mathcal{T}$, and use a following recursive procedure to partition $\mathtt{s}_0$. For each side-$\ell$ hypercube $\mathtt{s}$, there are 3 partitioning cases: (i) if $\mathtt{s}$ does not contain any data points, we discard it, (ii) if $\mathtt{s}$ contains 1 data point, we use the center of $\mathtt{s}$ (or the data point) as a node in $\mathcal{T}$, and (iii) if $\mathtt{s}$ contains more than 1 data point, we represent $\mathtt{s}$ by its center as a node $x$ in $\mathcal{T}$, and equally partition $\mathtt{s}$ into $2^{\mathtt{d}}$ side-$(\ell/2)$ hypercubes for potential child nodes of $x$. We then apply the recursive partition procedure for those child hypercubes. One can use any metrics in $\mathbb{R}^{\mathtt{d}}$ to obtain lengths for edges in $\mathcal{T}$. Additionally, one can use a predefined deepest level of $\mathcal{T}$ as a stopping condition for the procedure. We summarize the recursive tree construction procedure in Algorithm 1. As desired, the random expansion of the original hypercube into $\mathtt{s}_0$ creates a variety to partition data spaces. Note that Algorithm 1 for constructing tree $\mathcal{T}$ is also known as the classical `Quadtree` algorithm [56] for 2-dimensional data (and later extended for high-dimensional data in [6, 30, 31, 32]).

• **Clustering-based tree metrics.** As in Algorithm 1, the number of partitioned hypercubes grows exponentially with respect to dimension $\mathtt{d}$. To overcome this problem for high-dimensional spaces, we directly leverage the distribution of support data points to adaptively partition data spaces via clustering, inspired by the clustering-based approach for a space subdivision in Improved Fast Gauss Transform [48, 66]. We derive a similar recursive procedure as in the partition-based tree metrics, but apply the farthest-point clustering [27] to partition support data points, and replace centers of hypercubes by cluster centroids as nodes in $\mathcal{T}$. In practice, we fix the same number of clusters $\kappa$ when performing the farthest-point clustering (replace the partition in line 9 in Algorithm 1). $\kappa$ is typically chosen via cross-validation. In general, one can apply any favorite clustering methods. We use the farthest-point clustering due to its fast computation. In particular, the complexity of the farthest-point clustering into $\kappa$ clusters for $n$ data points is $O(n \log \kappa)$ using the algorithm in [23]. Using different random initializations for the farthest-point clustering, we recover a simple sampling mechanism to obtain random tree metrics.

## 5    Relations to Other Work

**OT with ground ultrametrics.** An ultrametric is also known as non-Archimedean metric, or isosceles metric [59]. Ultrametrics strengthen the triangle inequality to a strong inequality (i.e., for any $x, y, z$ in an ultrametric space, $d(x, z) \leq \max(d(x, y), d(y, z))$). Note that binary metrics are a special case of ultrametrics since binary metrics satisfy the strong inequality. Following [33, §1, p.245–247], an ultrametric implies a tree structure which can be constructed by hierarchical clustering schemes. Therefore, an ultrametric is a tree metric. Furthermore, we note that ultrametrics have similar spirits with strong kernels and hierarchy-induced kernels which are key components to form valid optimal assignment kernels for graph classification applications [37].

**Connection with OT with Euclidean ground metric $W_2(\cdot, \cdot)$.** Let $d_{\mathcal{T}}^H$ be a partition-based tree metric where $H$ is the depth level of corresponding tree $\mathcal{T}$, at which all support data points are separated into different hypercubes (i.e., Algorithm 1 stops at depth level $H$). Edges in $\mathcal{T}$ are computed by Euclidean distance. Let $\beta$ be the side of the randomly expanded hypercube. Given two $\mathtt{d}$-dimensional point clouds $\tilde{\mu}, \tilde{\nu}$ with the same cardinality (i.e., discrete uniform measures), and denote TW with $d_{\mathcal{T}}^H$ as $W_{d_{\mathcal{T}}^H}$. Then,

$$W_2(\tilde{\mu}, \tilde{\nu}) \leq W_{d_{\mathcal{T}}^H}(\tilde{\mu}, \tilde{\nu})/2 + \beta\sqrt{\mathtt{d}}/2^H.$$

The proof is given in the supplementary material. Moreover, we also investigate the empirical relation between the TSW distance and the $W_2$ distance in the supplementary material, in which empirical results indicate that the TSW distance agrees more with $W_2$ as the number of tree-slices used to define the TSW distance is increased.

**Connection with embedding $W_2$ metric into $\ell_1$ metric for fast nearest neighbor search.** As discussed earlier, our goal is neither to approximate OT distance using trees as in [7, 8, 13, 22, 30], nor to embed $W_2$ metric into $\ell_1$ metric as in [16, 32], but rather to sample tree metrics to define an extended variant of the sliced-Wasserstein distance. When using the `Quadtree` algorithm (as in Algorithm 1) to sample tree metrics for the TSW distance, then the resulted TSW distance is in the same spirit as the embedding approach in [32] where the authors embedded $W_2$ metric into $\ell_1$ metric by using a series of grids.

**OT with tree metrics.** There are a few work related to our considered class of OT with tree metrics [35, 62]. In particular, Kloeckner [35] studied geometric properties of OT space for measures on an ultrametric space, and Sommerfeld and Munk [62] focused on statistical inference for empirical OT on finite spaces including tree metrics.

## 6 Experimental Results

In this section, we evaluated the proposed TSW kernel $k_{\mathrm{TSW}}$ (Equation (5)) for comparing empirical measures in word embedding-based document classification and topological data analysis.

### 6.1 Word Embedding-based Document Classification

Kusner et al. [39] proposed Word Mover's distances for document classification. Each document is regarded as an empirical measure where each word and its frequency are considered as a support and a corresponding weight respectively. Kusner et al. [39] used word embedding such as *word2vec* to map each word to a vector data point. Equivalently, Word Mover's distances are OT metrics between empirical measures (i.e., documents) where its ground cost is a metric on the word embedding space.

**Setup.** We evaluated $k_{\mathrm{TSW}}$ on four datasets: TWITTER, RECIPE, CLASSIC and AMAZON, following the approach of Word Mover's distances [39], for document classification with SVM. Statistical characteristics for those datasets are summarized in Figure 2b. We used the *word2vec* word embedding [47], pre-trained on Google News[4], containing about 3 million words/phrases. *word2vec* maps these words/phrases into vectors in $\mathbb{R}^{300}$. Following [39], for all datasets, we removed all SMART stop word [55], and further dropped words in documents if they are not available in the pre-trained *word2vec*. We used two baseline kernels in the form of $\exp(-td)$ where $d$ is a document distance and $t > 0$, for two corresponding baseline document distances based on Word Mover's: (i) OT with Euclidean ground metric [39], and (ii) sliced-Wasserstein, denoted as $k_{\mathrm{OT}}$ and $k_{\mathrm{SW}}$ respectively. For TSW distance in $k_{\mathrm{TSW}}$, we consider $n_s$ randomized clustering-based tree metrics, built with a predefined deepest level $H_{\mathcal{T}}$ of tree $\mathcal{T}$ as a stopping condition. We also regularized for kernel $k_{\mathrm{OT}}$ matrices due to its indefiniteness by adding a sufficiently large diagonal term as in [14]. For SVM, we randomly split each dataset into $70\%/30\%$ for training and test with 100 repeats, choose hyper-parameters through cross validation, choose $1/t$ from $\{1, q_{10}, q_{20}, q_{50}\}$ where $q_s$ is the $s\%$ quantile of a subset of corresponding distances, observed on a training set, use one-vs-one strategy with Libsvm [12] for multi-class classification, and choose SVM regularization from $\{10^{-2:1:2}\}$. We ran experiments with Intel Xeon CPU E7-8891v3 (2.80GHz), and 256GB RAM.

### 6.2 Topological Data Analysis (TDA)

TDA has recently gained interest within the machine learning community [11, 38, 42, 53]. TDA is a powerful tool for statistical analysis on geometric structured data such as linked twist maps, or material data. TDA employs algebraic topology methods, such as persistence homology, to extract robust topological features (i.e., connected components, rings, cavities) and output 2-dimensional point multisets, known as persistence diagrams (PD) [19]. Each 2-dimensional point in PD summarizes a lifespan, corresponding to birth and death time as its coordinates, of a particular topological feature.

**Setup.** We evaluated $k_{\mathrm{TSW}}$ for orbit recognition and object shape classification with support vector machines (SVM), as well as change point detection for material data analysis with kernel Fisher discriminant ratio (KFDR) [28]. Generally, we followed the same setting as in [42] for these TDA experiments. We considered five baseline kernels for PD: (i) persistence scale space ($k_{\mathrm{PSS}}$) [53], (ii) persistence weighted Gaussian ($k_{\mathrm{PWG}}$) [38], (iii) sliced-Wasserstein ($k_{\mathrm{SW}}$) [11], (iv) persistence Fisher ($k_{\mathrm{PF}}$) [42], and (v) optimal transport[5], defined as $k_{\mathrm{OT}} = \exp(-td_{\mathrm{OT}})$ for $t > 0$, and also further regularized its kernel matrices by adding a sufficiently large diagonal term due to its indefiniteness as in §6.1. For TSW distance in $k_{\mathrm{TSW}}$, we considered $n_s$ randomized partition-based tree metrics, built with a predefined deepest level $H_{\mathcal{T}}$ of tree $\mathcal{T}$ as a stopping condition.

Let $\mathrm{Dg}_i = (x_1, x_2, \ldots, x_n)$ and $\mathrm{Dg}_j = (z_1, z_2, \ldots, z_m)$ be two PD where $x_i \mid_{1 \le i \le n}, z_j \mid_{1 \le j \le m} \in \mathbb{R}^2$, and $\Theta = \{(a, a) \mid a \in \mathbb{R}\}$ be the diagonal set. Denote $\mathrm{Dg}_{i\Theta} = \{\Pi_\Theta(x) \mid x \in \mathrm{Dg}_i\}$ where $\Pi_\Theta(x)$ is a projection of $x$ on $\Theta$. As in SW distance between $\mathrm{Dg}_i$ and $\mathrm{Dg}_j$ [11], we use transportation plans between $(\mathrm{Dg}_i \cup \mathrm{Dg}_{j\Theta})$ and $(\mathrm{Dg}_j \cup \mathrm{Dg}_{i\Theta})$ for TW (in Equation (4) of TSW) and OT distances. We typically used a cross validation to choose hyper-parameters, and followed corresponding authors of those baseline kernels to form sets of candidates. For $k_{\mathrm{TSW}}$ and $k_{\mathrm{OT}}$, we chose $1/t$ from $\{1, q_{10}, q_{20}, q_{50}\}$. Similar as in §6.1, we used one-vs-one strategy with Libsvm for multi-class classification, $\{10^{-2:1:2}\}$ as a set of regularization candidates, and a random split $70\%/30\%$ for training and test with 100 repeats for SVM, and DIPHA toolbox[6] to extract PD.

**Orbit recognition.**  Adams et al. [1, §6.4.1] proposed a synthesized dataset for link twist map, a discrete dynamical system to model flows in DNA microarrays [29]. There are 5 classes of orbits. As in [42], we generated 1000 orbits for each class where each orbit contains 1000 points. We considered 1-dimensional topological features for PD, extracted with Vietoris-Rips complex filtration [19].

**Object shape classification.**  We evaluated object shape classification on a 10-class subset of MPEG7 dataset [40], containing 20 samples for each class as in [42]. For simplicity, we used the same procedure as in [42] to extract 1-dimensional topological features for PD with Vietoris-Rips complex filtration[7] [19].

**Change point detection for material data analysis.**  We considered granular packing system [24] and $SiO_2$ [49] datasets for change point detection problem with KFDR as a statistical score. As in [38, 42], we extracted 2-dimensional topological features for PD in granular packing system dataset, 1-dimensional topological features for PD in $SiO_2$ dataset, both with ball model filtration, and set $10^{-3}$ for the regularization parameter in KFDR. KFDR graphs for these datasets are shown in Figure 2c. For granular tracking system dataset, all kernel approaches obtain the change point as the $23^{rd}$ index, which support an observation result (corresponding id = 23) in [5] . For $SiO_2$ dataset, results of all kernel methods are within a supported range ($35 \le \mathrm{id} \le 50$), obtained by a traditional physical approach [20]. The KFDR results of $k_{\mathrm{TSW}}$ compare favorably with those of other baseline kernels. As shown in Figure 2b, $k_{\mathrm{TSW}}$ is faster than other baseline kernels. We note that we omit the baseline kernel $k_{\mathrm{OT}}$ for this application since computation of OT distance is out of memory.

### 6.3   Results of SVM, Time Consumption and Discussion

The results of SVM and time consumption for kernel matrices in TDA, and word embedding based document classification are illustrated in Figure 2a and Figure 2b respectively. The performances of $k_{\mathrm{TSW}}$ compare favorably with other baseline kernels. Moreover, the computational time of $k_{\mathrm{TSW}}$ is much less than that of $k_{\mathrm{OT}}$. Especially, in CLASSIC dataset, it took less than 3 hours for $k_{\mathrm{TSW}}$ while more than 8 days for $k_{\mathrm{OT}}$. Note that $k_{\mathrm{TSW}}$ and $k_{\mathrm{SW}}$ are positive definite while $k_{\mathrm{OT}}$ is not. The indefiniteness of $k_{\mathrm{OT}}$ may affect its performances in some applications, e.g. $k_{\mathrm{OT}}$ performs worse in TDA applications, but works well for documents with word embedding applications. The fact that SW only considers 1-dimensional projections may limit its ability to capture high-dimensional structure in data distributions [60]. TSW distance remedies this problem by using clustering-based tree metrics which directly leverage distributions of support data points. Furthermore, we also illustrate a trade-off of performances and computational time for different parameters in tree-sliced-Wasserstein distances for $k_{\mathrm{TSW}}$ on TWITTER dataset in Figure 2d. For tree-sliced-Wasserstein TSW for $k_{\mathrm{TSW}}$, performances are usually improved with more slices ($n_s$), but they come with a trade-off of more computational time. In these applications, we observed that a good trade-off for $n_s$ of tree-sliced-Wasserstein is about 10 slices. Many further results can be seen in the supplementary.

## 7   Conclusion

In this work, we proposed positive definite tree-(sliced)-Wasserstein kernel on OT geometry by considering a particular class of ground metrics, namely tree metrics. Much like the univariate Wasserstein distance, the tree-(sliced)-Wasserstein distance has a closed form, and is also negative

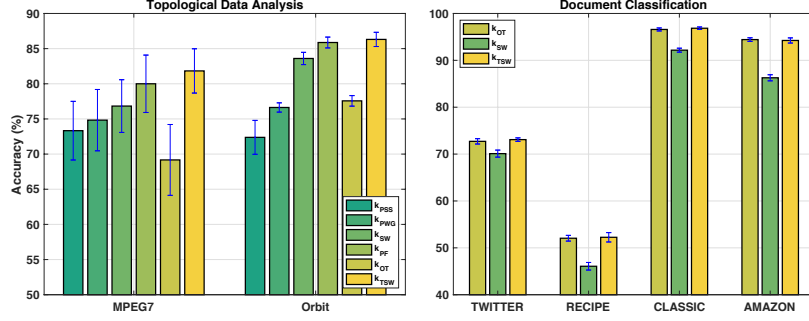

(a) SVM results for TDA and document classification.

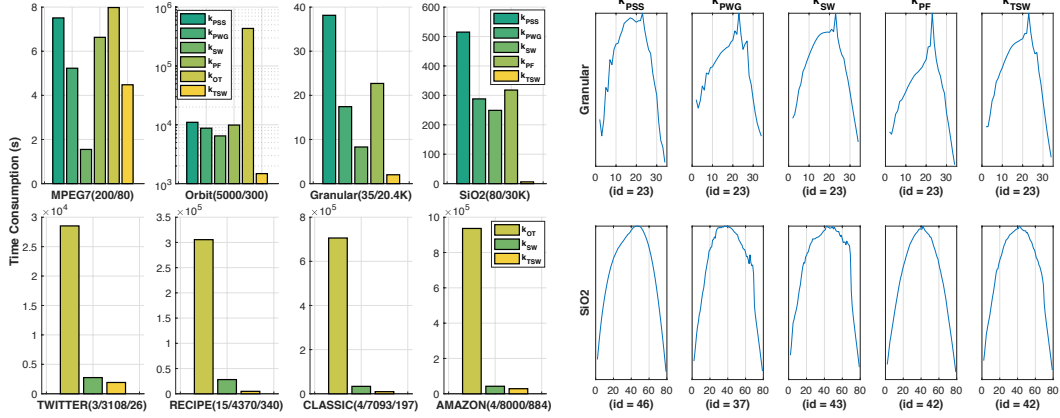

(b) Corresponding time consumption of kernel matrices for TDA and document classification.

(c) The KFDR graphs on granular packing system and SiO₂ datasets.

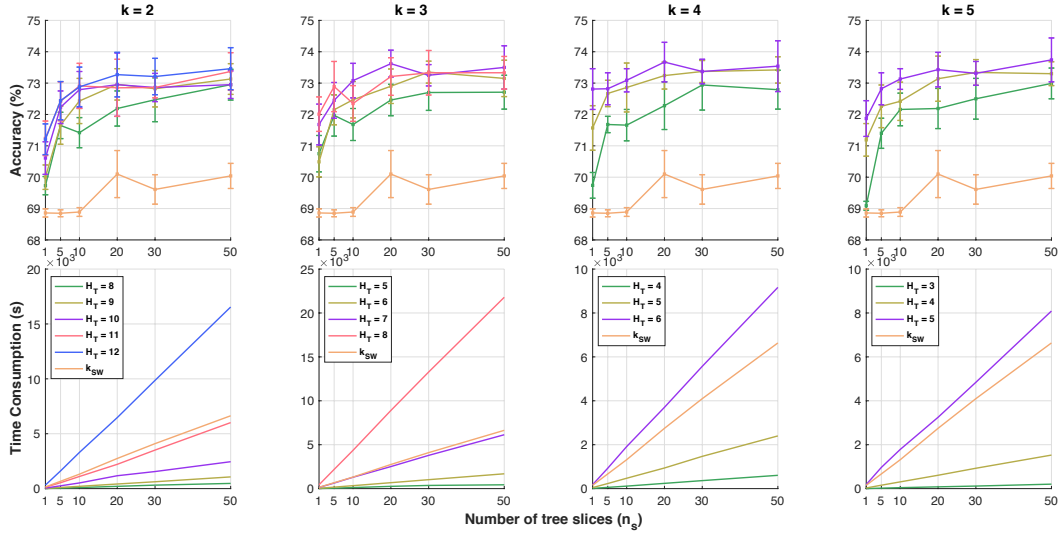

(d) SVM results and time consumption of kernel matrices of $k_{\text{TSW}}$ with different $(n_s, H_{\mathcal{T}}, \kappa)$, and $k_{\text{SW}}$ with different $n_s$ on TWITTER dataset.

Figure 2: Experimental results for document classification and TDA. In (a), for a trade-off between time consumption and performances, results of TDA are reported for $k_{\text{TSW}}$ with $(n_s = 6, H_{\mathcal{T}} = 6)$, and $(n_s = 12, H_{\mathcal{T}} = 5)$ in MPEG7 and Orbit datasets respectively. For document classification, results are reported for $k_{\text{SW}}$ with $(n_s = 20)$, and for $k_{\text{TSW}}$ with $(n_s = 10, H_{\mathcal{T}} = 6, \kappa = 4)$. In (b), the numbers in the parenthesis: for TDA in the first row, are the number of PD and the maximum number of points in PD respectively; for document classification in the second row, are the number of classes, the number of documents, and the maximum number of unique words for each document respectively. In (c), for $k_{\text{TSW}}$, TSW distances are computed with $(n_s = 12, H_{\mathcal{T}} = 6)$.

definite. We also provide two sampling schemes to generate tree metrics for both high-dimensional and low-dimensional spaces. Leveraging random tree-metrics, we have proposed a new generalization of sliced-Wasserstein metrics that has more flexibility and degrees of freedom, by choosing a tree rather than a line, especially in high-dimensional spaces. The questions of sampling efficiently tree metrics from data points for tree-sliced-Wasserstein distance, as well as using them for more involved parametric inference are left for future work.

**Acknowledgments**

We thank anonymous reviewers for their comments. TL acknowledges the support of JSPS KAKENHI Grant number 17K12745. MY was supported by the JST PRESTO program JPMJPR165A.

## Footnotes

[1]In general, Wasserstein spaces are not Hilbertian [52, §8.3].

[2]https://github.com/lttam/TreeWasserstein.

[3]We follow here [9, p. 66–67], to define negative-definiteness, see review about kernels in the supplementary.

[4]https://code.google.com/p/word2vec

[5]We used a fast OT implementation (e.g. on MPEG7 dataset, it took 7.98 seconds while the popular mex-file with Rubner's implementation required 28.72 seconds).

[6]https://github.com/DIPHA/dipha

[7]Turner et al. [63] proposed a more complicated and advanced filtration for this task.

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
