[Supplementary Material]

# Supplementary Material for: Tree-Sliced Variants of Wasserstein Distances

**Tam Le**
RIKEN AIP, Japan
tam.le@riken.jp

**Makoto Yamada**
Kyoto University & RIKEN AIP, Japan
makoto.yamada@riken.jp

**Kenji Fukumizu**
ISM, Japan & RIKEN AIP, Japan
fukumizu@ism.ac.jp

**Marco Cuturi**
Google Brain, Paris & CREST - ENSAE
cuturi@google.com

## 1 Detailed Proofs

In this section, we give detailed proofs for the inequality in the connection with OT with Euclidean ground metric (i.e. $W_2$ metric) for TW distance, and investigate an empirical relation between TSW and $W_2$ metric, especially when one increases the number of tree-slices in TSW. Additionally, we also provide proofs for negative definiteness of $\ell_1$ distance (used in the proof of Proposition 2 in the main text [Le et al., 2019c]), and indefinite divisibility for TSW kernel.

### 1.1 Proof of: $W_2(\tilde{\mu}, \tilde{\nu}) \le W_{d_{\mathcal{T}}^H}(\tilde{\mu}, \tilde{\nu})/2 + \beta\sqrt{\mathrm{d}}/2^H$

For two point clouds $\tilde{\mu}, \tilde{\nu}$ containing $n$ data points $x_i \mid_{1 \le i \le n}, z_j \mid_{1 \le j \le n}$ respectively, let $c$ be a ground cost metric, and $\Sigma_n$ be the set of all permutations of $n$ elements, the OT can be reformulated as an optimal assignment problem as follow:

$$W_c(\tilde{\mu}, \tilde{\nu}) = \min_{\sigma \in \Sigma_n} \frac{1}{n} \sum_{i=1}^{n} c(x_i, z_{\sigma(i)}). \tag{1}$$

At a height level $i$ in $\mathcal{T}$, the maximum Euclidean distance between any two data points in a same hypercube, denoted as $\Delta_i$, we have

$$\Delta_i = \beta\sqrt{\mathrm{d}}/2^i.$$

Let $E_{i(i+1)}$ be a set of all edges between a height level $i$ and a height level $(i + 1)$ in $\mathcal{T}$. So, for any $e \in E_{i(i+1)}$, we have

$$w_e = \beta\sqrt{\mathrm{d}}/2^{i+1}.$$

Let $q_i$ be the number of matched pairs at a height level $i$. Consequently, $(n - q_i)$ is the number of unmatched pairs at the height level $i$. Moreover, for the number of unmatched pairs at the height level $i$, observe that

$$n - q_i = \frac{1}{2} \sum_{e \in E_{(i-1)i}} |\tilde{\mu}(\Gamma(v_e)) - \tilde{\nu}(\Gamma(v_e))|. \tag{2}$$

In the right hand side of Equation (2), for all edges between the height level $(i-1)$ and $i$ of tree $\mathcal{T}$. Note that the total mass in subtree $\Gamma(v_e)$ is equal to the total mass flowing through edge $e$, we have $\mathcal{P}(r, x) \mid_{x \in \tilde{\mu}}$ and $\mathcal{P}(r, z) \mid_{z \in \tilde{\nu}}$ count the total number of visits on each edge in $E_{i(i+1)}$ for $\tilde{\mu}$ and $\tilde{\nu}$ respectively, and their absolute different number is twice to the number of unmatched pairs at the height level $i$, as described in Equation (2).

Figure 1: A frequency of near-neighbor rank on the $W_2$ space for the nearest neighbor w.r.t. TSW.

Therefore, the TW distance is as follow:

$$W_{d_{\mathcal{T}}^H}(\tilde{\mu}, \tilde{\nu}) = \frac{1}{n} \sum_{i=0}^{H-1} 2w_{\bar{e}_{i(i+1)}}(n - q_{i+1}) = \frac{1}{n} \sum_{i=0}^{H-1} \Delta_i(n - q_{i+1}), \tag{3}$$

where $\bar{e}_{i(i+1)}$ is an edge in $E_{i(i+1)}$, and note that $\Delta_i = 2w_{\bar{e}_{i(i+1)}}$.

Moreover, we have $(q_i - q_{i+1})$ is the number of pairs matched at a height level $i$, but unmatched at a height level $(i + 1)$. Additionally, note that $q_0 = n$, $q_H = 0$, and $\Delta_i = \Delta_{i-1}/2$, then we have

$$W_2(\tilde{\mu}, \tilde{\nu}) \le \frac{1}{n} \sum_{i=0}^{H-1} \Delta_i (q_i - q_{i+1}) \tag{4}$$

$$= W_{d_{\mathcal{T}}^H}(\tilde{\mu}, \tilde{\nu}) - \frac{1}{n} \sum_{i=0}^{H-1} \Delta_i(n - q_i) \tag{5}$$

$$= W_{d_{\mathcal{T}}^H}(\tilde{\mu}, \tilde{\nu}) - \frac{1}{n} \sum_{i=1}^{H} \Delta_{i-1}(n - q_i)/2 + \Delta_H \tag{6}$$

$$= W_{d_{\mathcal{T}}^H}(\tilde{\mu}, \tilde{\nu})/2 + \beta\sqrt{\mathrm{d}}/2^H. \tag{7}$$

For the first equal, we added and subtracted $n$ for the term in the parenthesis and note Equation (3) for $W_{d_{\mathcal{T}}^H}$. For the second equal, in the second term, for the element with $i = 0$, note that $(n - q_0) = 0$, we added and subtracted the element with $i = H$. For the third equal, we grouped the first two terms and note that $\Delta_H = \beta\sqrt{\mathrm{d}}/2^H$ for the third term.

**Empirical relation between TSW and OT with Euclidean ground metric.** The hypercube tree-sliced metric (i.e. partition-based tree metric) is our suggestion to build practical tree metrics for TSW when used on low-dimensional data spaces. We emphasize that we do not try to mimic the Euclidean OT (i.e. $W_2$) or the sliced-Wasserstein (SW), but rather propose a variant of OT distance. As stated in the main text [Le et al., 2019c], SW is a special case of TSW. From an empirical point of view, we have carried out the following experiment to investigate an empirical relation between TSW and $W_2$ distance:

For a query point $q$, let $p$ be its nearest neighbor w.r.t. TSW. Figure 1 illustrates that $p$ is very likely among the top 5 on MPEG7 dataset, and top 10 on Orbit dataset, near neighbors on the $W_2$ space. Results are averaged over 1000 runs of random split 90%/10% for training and test. When the number of tree-slices in TSW increases, the $W_2$ near-neighbor rank of $p$ is improved. These empirical results suggest that TSW may agree with some aspects of $W_2$.

Figure 2: Results of SVM and time consumption of kernel matrices of $k_{\text{TW}}$ with different $(n_s, H_{\mathcal{T}}, \kappa)$, and $k_{\text{SW}}$ with different $n_s$ on RECIPE dataset.

## 1.2 Proof of: Negative Definiteness for $\ell_1$ Distance

For two real numbers $a, b$, the function $(a, b) \mapsto (a - b)^2$ is obviously negative definite. Following [Berg et al., 1984, Corollary 2.10, p.78], the function $(a, b) \mapsto |a - b|$ is negative definite. Therefore, $\ell_1$ distance is a sum of negative definite functions. Thus, $\ell_1$ distance is negative definite.

## 1.3 Indefinite Divisibility for Tree-Sliced-Wasserstein Kernel

Inspired by Le and Yamada Le and Yamada [2018], we derive the following proof of indefinite divisibility for the TSW kernel. For probability $\mu, \nu$ on tree $\mathcal{T}$, and $i \in \mathbb{N}^*$, let $k_{\text{TSW}_i}(\mu, \nu) = \exp(-\frac{t}{i}\text{TSW}(\mu, \nu))$. We have $k_{\text{TSW}}(\cdot, \cdot) = (k_{\text{TSW}_i}(\cdot, \cdot))^i$, and $k_{\text{TSW}_i}(\cdot, \cdot)$ is positive definite. Following [Berg et al., 1984, §3, Definition 2.6, p.76], $k_{\text{TSW}}$ is indefinitely divisible. Therefore, one does not need to recompute the Gram matrix of TSW kernel for each choice of $t$, since it indeed suffices to compute TSW distances between empirical measures in a training set once.

## 2 More Experimental Results

We provide many further experimental results for our proposed tree-Wasserstein kernel on word embedding-based document classification and topological data analysis (TDA).

### 2.1 Word Embedding-based Document Classification

Figure 2, Figure 3, and Figure 4 show SVM results and time consumption of kernel matrices of $k_{\text{TW}}$ with different $(n_s, H_{\mathcal{T}}, \kappa)$, and $k_{\text{SW}}$ with different $n_s$ on RECIPE, CLASSIC, and AMAZON datasets respectively.

### 2.2 Topological Data Analysis

**Orbit recognition.** Figure 5 shows SVM results and time consumption of kernel matrices of $k_{\text{TW}}$ with different $(n_s, H_{\mathcal{T}})$ on Orbit dataset.

Figure 3: Results of SVM and time consumption of kernel matrices of $k_{\mathrm{TW}}$ with different $(n_s, H_{\mathcal{T}}, \kappa)$, and $k_{\mathrm{SW}}$ with different $n_s$ on CLASSIC dataset.

Figure 4: Results of SVM and time consumption of kernel matrices of $k_{\mathrm{TW}}$ with different $(n_s, H_{\mathcal{T}}, \kappa)$, and $k_{\mathrm{SW}}$ with different $n_s$ on AMAZON dataset.

Figure 5: Results of SVM and time consumption of kernel matrices of $k_{\mathrm{TW}}$ with different $(n_s, H_{\mathcal{T}})$ on Orbit dataset.

Figure 6: Results of SVM and time consumption of kernel matrices of $k_{\mathrm{TW}}$ with different $(n_s, H_{\mathcal{T}})$ on MPEG7 dataset.

**Object shape classification.** Figure 6 shows SVM results and time consumption of kernel matrices of $k_{\mathrm{TW}}$ with different $(n_s, H_{\mathcal{T}})$ on MPEG7 dataset.

**Change point detection for material data analysis.** Figure 7 and Figure 8 show time consumption of kernel matrices of $k_{\mathrm{TW}}$ with different $(n_s, H_{\mathcal{T}})$ on granular packing system and $SiO_2$ datasets respectively.

## 3 Some Other Relations to Other Work

**OT with linear chain tree metrics.** For a metric $d$ in 1-dimensional spaces of supports, all support data points lay on a line which is a trivial case of a tree .Therefore, all data points are nodes in a tree, and a length of an edge equals to the distance $d$ between two nodes of the edge. Thus, $d$ is a tree metric. Moreover, one can generalize the metric $d$ in 1-dimensional spaces of supports into a geodesic distance $d'$ of 1-dimensional curved manifolds, as considered in [Kolouri et al., 2019]. Similarly, one can construct a tree $\mathcal{T}$ along the 1-dimensional curved manifolds where all data points are nodes in $\mathcal{T}$, and lengths of edges are computed by geodesic distance $d'$. Therefore, $d'$ is also a tree metric.

**Connection with sliced-Wasserstein distances.** SW is a popular variant of OT [Rabin et al., 2011, Bonneel et al., 2015]. SW exploits a closed form computation of OT for 1-dimensional spaces of supports by working directly with projected support data points on a real line. Since OT with ground metrics in 1-dimensional spaces of supports is a special case of TW distance, SW distance is

Figure 7: Time consumption of kernel matrices of $k_{\mathrm{TW}}$ with different $(n_s, H_{\mathcal{T}})$ on granular packing system dataset.

Figure 8: Time consumption of kernel matrices of $k_{\mathrm{TW}}$ with different $(n_s, H_{\mathcal{T}})$ on SiO$_2$ dataset.

consequently a special case of tree-sliced-Wasserstein distance. Thus, tree-sliced-Wasserstein not only preserves merits of SW, but provides more flexibility since choosing a tree rather than a line has far more degrees of freedom, especially in high-dimensional spaces of supports.

**Positive definite kernels on OT geometry.** Besides the sliced-Wasserstein kernel [Kolouri et al., 2016, Carriere et al., 2017] which is a special case of our proposed kernel, as far as we know, there are only the permanent [Cuturi et al., 2007] and generating function [Cuturi, 2012] kernels. However, they are intractable.

**OT with tree metrics.** Recently, Le et al. also leveraged the structure of tree metrics to develop scalable tree variants of Wassserstein barycenter [Le et al., 2019b] and Gromov-Wasserstein [Le et al., 2019a] for large-scale applications.

# 4 Brief Reviews of Kernels, the Farthest-Point Clustering, and the Synthesized Orbit Dataset

In this section, we give brief reviews for kernels, and the farthest-point clustering Gonzalez [1985].Then, we provide details for the synthesized orbit dataset for orbit recognition).

### 4.1 A Brief Review of Kernels

We review some important definitions and theorems about kernels used in our work.

**Positive definite kernels [Berg et al., 1984, p.66–67].** A kernel function $k : \mathcal{X} \times \mathcal{X} \to \mathbb{R}$ is positive definite if $\forall n \in \mathbb{N}^*, \forall x_1, x_2, ..., x_n \in \mathcal{X}, \sum_{i,j} c_i c_j k(x_i, x_j) \geq 0, \forall c_i \in \mathbb{R}$.

**Negative definite kernels [Berg et al., 1984, p.66–67].** A kernel function $k : \mathcal{X} \times \mathcal{X} \to \mathbb{R}$ is negative definite if $\forall n \geq 2, \forall x_1, x_2, ..., x_n \in \mathcal{X}, \sum_{i,j} c_i c_j k(x_i, x_j) \leq 0, \forall c_i \in \mathbb{R}$ such that $\sum_i c_i = 0$.

**Berg-Christensen-Ressel Theorem [Berg et al., 1984, Theorem 3.2.2, p.74].** If $\kappa$ is a *negative definite* kernel, then kernel $k_t(x, z) := \exp(-t\kappa(x, z))$ is positive definite for all $t > 0$.

### 4.2 A Brief Review of the Farthest-Point Clustering for Data Space Partition

The data space partition can be modeled as a $\kappa$-center problem. Given $n$ data points $x_1, x_2, ..., x_n$, and a predefined number of clusters $\kappa$, the goal of $\kappa$-center problem is to find a partition of $n$ points into $\kappa$ clusters $\mathtt{S}_1, \mathtt{S}_2, ..., \mathtt{S}_\kappa$ as well as their corresponding centers $\mathtt{c}_1, \mathtt{c}_2, ..., \mathtt{c}_\kappa$ to minimize the maximum radius of clusters.

The farthest-point clustering Gonzalez [1985] is a simple greedy algorithm, summarized in Algorithm 1. Gonzalez Gonzalez [1985] also proved that the farthest-point clustering computes a partition with maximum radius at most twice the optimum for $\kappa$-center clustering. The complexity for a direct implementation for the farthest-point clustering as in Algorithm 1 is $O(n\kappa)$. This complexity can be reduced into $O(n \log \kappa)$ by using the algorithm in Feder and Greene [1988].

---

**Algorithm 1** `Farthest_Point_Clustering`$(X, \kappa)$

---

**Input:** $X = (x_1, x_2, \ldots, x_n)$: a set of $n$ data points, and $\kappa$: the predefined number of clusters.
**Output:** Clustering centers $\mathtt{c}_1, \mathtt{c}_2, ..., \mathtt{c}_\kappa$ and cluster index for each point $x_i$.
1: $\mathtt{c}_1 \leftarrow$ a random point $x \in X$.
2: Set of cluster $C \leftarrow \mathtt{c}_1$.
3: $i \leftarrow 1$.
4: **while** $i < \kappa$ and $n - i > 0$ **do**
5:    $i \leftarrow i + 1$.
6:    $\mathtt{c}_i \leftarrow \max_{x_j \in X} \min_{\mathtt{c} \in C} \|x_j - \mathtt{c}\|$. (a farthest point $x_j \in X$ to $C$).
7:    $C \leftarrow C \cup \mathtt{c}_i$. (Add the new center into $C$).
8: **end while**
9: Each data point $x_j \in X$ is assigned to its nearest center $\mathtt{c}_i \in C$.

---

### 4.3 Details of the Synthesized Orbit Dataset

Adams et al. [Adams et al., 2017, §6.4.1] proposed a synthesized dataset for link twist map, a discrete dynamical system to model flows in DNA microarrays Hertzsch et al. [2007].

Given an initial position $(a_0, b_0) \in [0, 1]^2$, and $t > 0$, an orbit is modeled as

$$a_{i+1} = \quad a_i + t b_i(1 - b_i) \mod 1, \tag{8}$$
$$b_{i+1} = \quad b_i + t a_{i+1}(1 - a_{i+1}) \mod 1. \tag{9}$$

There are 5 classes, corresponding to 5 different parameters $t = 2.5, 3.5, 4, 4.1, 4.3$. For each class, we generated 1000 orbits with random initial positions where each orbit contains 1000 points.

## 5 More Examples on the Partition-based Tree Metric, Quantization and Cluster Sensitivity Problems, and Persistence Diagrams

In this section, we give some examples for the partition-based tree metric, quantization and cluster sensitivity problems and persistence diagrams.

Figure 9: An example about the partition-based tree metric for a set of points in a 2-dimensional space.

Figure 10: The corresponding tree structure for the example in Figure 9.

## 5.1 An Example on the Partition-based Tree Metric

Given a set $X$ of 7 data points $x_i \mid_{1 \leq i \leq 7}$ in a 2-dimensional space as illustrated in Figure 9, one can choose a square region $\mathtt{s}_0$ as the largest square in Figure 9 containing all data points in $X$, and denote $\ell$ as the side of the largest square.

At height level 0 in tree $\mathcal{T}$, applying the `Partition_HC` algorithm for $\mathtt{s}_0$, one has $x_{\Delta_1}$ (center of $\mathtt{s}_0$) as a node (i.e., the root) represented for $\mathtt{s}_0$ in the constructed tree structure $\mathcal{T}$, and 4 child square regions[1] with side $\ell/2$, denoted $\mathtt{s}_{1a}$ (containing $x_1$), $\mathtt{s}_{1b}$ (containing $x_2, x_3, x_4, x_5, x_7$), $\mathtt{s}_{1c}$ (containing $x_6$), and $\mathtt{s}_{1d}$ (containing no data points). Therefore, one can discard $\mathtt{s}_{1d}$, use either data points ($x_1$, or $x_6$) or their centers represented for $\mathtt{s}_{1a}$ and $\mathtt{s}_{1c}$ respectively, and then apply the recursive procedure to partition for $\mathtt{s}_{1b}$ (at height level 1).

At height level 1 in tree $\mathcal{T}$, applying the `Partition_HC` algorithm for $\mathtt{s}_{1b}$, one has $x_{\Delta_2}$ (center of $\mathtt{s}_{1b}$) as a node represented for $\mathtt{s}_{1b}$ in the constructed tree structure $\mathcal{T}$, and 4 child square regions with side $\ell/4$, denoted $\mathtt{s}_{2a}$ (containing $x_3, x_4, x_5$), $\mathtt{s}_{2b}$ (containing $x_7$), $\mathtt{s}_{2c}$ (containing no data points), and $\mathtt{s}_{2d}$ (containing $x_2$). Therefore, one can discard $\mathtt{s}_{2c}$, use either data points ($x_7$, or $x_2$) or their centers represented for $\mathtt{s}_{2b}$ and $\mathtt{s}_{2d}$ respectively, and then apply the recursive procedure to partition for $\mathtt{s}_{2a}$ (at height level 2).

At height level 2 in tree $\mathcal{T}$, applying the `Partition_HC` algorithm for $\mathtt{s}_{2a}$, one has $x_{\Delta_3}$ (center of $\mathtt{s}_{2a}$) as a node represented for $\mathtt{s}_{2a}$ in the constructed tree structure $\mathcal{T}$, and 4 child square regions with side $\ell/8$, denoted $\mathtt{s}_{3a}$ (containing no data points), $\mathtt{s}_{3b}$ (containing $x_5$), $\mathtt{s}_{3c}$ (containing $x_4$), and $\mathtt{s}_{3d}$ (containing $x_3$). Therefore, one can discard $\mathtt{s}_{3a}$, and use either data points ($x_5$, or $x_4$, or $x_3$) or their centers represented for $\mathtt{s}_{3b}$, $\mathtt{s}_{3c}$ and $\mathtt{s}_{3d}$ respectively.

Hence, at the end, one obtains a tree structure $\mathcal{T}$ as illustrated in Figure 10, containing 10 nodes $v_i \mid_{1 \leq i \leq 10}$, and 9 edges $e_j \mid_{1 \leq j \leq 9}$. Node $x_1$ is the root of $\mathcal{T}$. The highest level in tree $\mathcal{T}$ is 3. For lengths of edges in $\mathcal{T}$, one can apply any metrics in the 2-dimensional space.

## 5.2 Some Examples and Discussion about the Quantization and Cluster Sensitivity Problems

The quantization or cluster sensitivity problem for partition or clustering is that some close data points are partitioned or clustered to adjacent, but different hypercubes or clusters respectively.

For example, in Figure 11, we illustrate different results of clustering, obtained with different initializations for the farthest-point clustering for a given 10000 random data points into 20 clusters. For data points near a border of adjacent, but different clusters, although they are very close to each other, they are still in different clusters, or known as a cluster sensitivity problem. Whether those data points are clustered into the same or different cluster(s), it depends on an initialization of the farthest-point clustering. Therefore, by combining many different clustering results, obtained with various initializations for the farthest-point clustering algorithm, one can reduce an affect of the cluster sensitivity problem. Similarly for a quantization problem in the partition procedure (e.g. those data points near a border of adjacent, but different square regions of the same side in Figure 9).

## 5.3 An Example of Persistence Diagrams

In Figure 12, we give an example of a persistence diagram on a real-value function $f : \mathcal{X} \mapsto \mathbb{R}$. Persistence homology considers a family of sublevel sets $f^{-1}((-\infty, t])$. When $t$ in $f^{-1}((-\infty, t])$ goes from $-\infty$ to $+\infty$, we collect all topological events, e.g., births and deaths of connected components (i.e., 0-dimensional topological features). As in Figure 12, connected components appears (i.e., birth) at $t = t_1, t_2$, and disappear (i.e., death) at $t = +\infty, t_3$ respectively. Therefore, persistence diagram of $f$ is $\mathrm{Dg}f = \{(t_1, +\infty), (t_2, t_3)\}$.

## Footnotes

[1] we use a clock order to enumerate for those child square regions: top right –> bottom right –> bottom left –> top left.

# References

Henry Adams, Tegan Emerson, Michael Kirby, Rachel Neville, Chris Peterson, Patrick Shipman, Sofya Chepushtanova, Eric Hanson, Francis Motta, and Lori Ziegelmeier. Persistence images: A

Figure 11: An illustration of the farthest-point clustering for 10000 data points into 20 clusters with different initializations.

Figure 12: An example of a persistence diagram on a real-value function $f : \mathcal{X} \mapsto \mathbb{R}$. With sublevel sets $f^{-1}((-\infty, t])$ filtration, persistence diagram of $f$ is $\mathrm{Dg}f = \{(t_1, +\infty), (t_2, t_3)\}$.

stable vector representation of persistent homology. *The Journal of Machine Learning Research*, 18(1):218–252, 2017.

Christian Berg, Jens Peter Reus Christensen, and Paul Ressel. *Harmonic analysis on semigroups*. Springer-Verlag, 1984.

Nicolas Bonneel, Julien Rabin, Gabriel Peyré, and Hanspeter Pfister. Sliced and Radon Wasserstein barycenters of measures. *Journal of Mathematical Imaging and Vision*, 51(1):22–45, 2015.

Mathieu Carriere, Marco Cuturi, and Steve Oudot. Sliced Wasserstein kernel for persistence diagrams. In *International Conference on Machine Learning*, volume 70, pages 664–673, 2017.

Marco Cuturi. Positivity and transportation. *arXiv preprint arXiv:1209.2655*, 2012.

Marco Cuturi, Jean-Philippe Vert, Oystein Birkenes, and Tomoko Matsui. A kernel for time series based on global alignments. In *Acoustics, Speech and Signal Processing (ICASSP)*, volume 2, pages II–413, 2007.

Tomas Feder and Daniel Greene. Optimal algorithms for approximate clustering. In *Proceedings of the twentieth annual ACM symposium on Theory of computing*, pages 434–444. ACM, 1988.

Teofilo F Gonzalez. Clustering to minimize the maximum intercluster distance. *Theoretical Computer Science*, 38:293–306, 1985.

Jan-Martin Hertzsch, Rob Sturman, and Stephen Wiggins. Dna microarrays: design principles for maximizing ergodic, chaotic mixing. *Small*, 3(2):202–218, 2007.

Soheil Kolouri, Yang Zou, and Gustavo K Rohde. Sliced Wasserstein kernels for probability distributions. In *Proceedings of the IEEE Conference on Computer Vision and Pattern Recognition (CVPR)*, pages 5258–5267, 2016.

Soheil Kolouri, Kimia Nadjahi, Umut Simsekli, Roland Badeau, and Gustavo K Rohde. Generalized sliced wasserstein distances. *arXiv preprint arXiv:1902.00434*, 2019.

Tam Le and Makoto Yamada. Persistence Fisher kernel: A Riemannian manifold kernel for persistence diagrams. In *Advances in Neural Information Processing Systems*, pages 10028–10039, 2018.

Tam Le, Nhat Ho, and Makoto Yamada. Computationally efficient tree variants of gromov-wasserstein. *arXiv preprint arXiv:1910.04462*, 2019a.

Tam Le, Viet Huynh, Nhat Ho, Dinh Phung, and Makoto Yamada. On scalable variant of wasserstein barycenter. *arXiv preprint arXiv:1910.04483*, 2019b.

Tam Le, Makoto Yamada, Kenji Fukumizu, and Marco Cuturi. Tree-sliced variants of wasserstein distances. In *Advances in neural information processing systems*, pages 12283–12294, 2019c.

Julien Rabin, Gabriel Peyré, Julie Delon, and Marc Bernot. Wasserstein barycenter and its application to texture mixing. In *International Conference on Scale Space and Variational Methods in Computer Vision*, pages 435–446, 2011.