[Reviews · NeurIPS 2019]

Reviewer 1



This paper describes a family of optimal transport metrics that are efficiently computable. In particular, these are 1-Wasserstein distances in which the ground metric is tree-structured. The paper gives a closed form for the distance whose computation scales as the number of edges in the tree. Several facts about this family of OT metrics are presented, including negative definiteness. Empirical results investigate a "tree-slicing" scheme, in which multiple trees are constructed and the resulting distances are averaged, showing favorable tradeoffs between computational cost and performance in several statistical tasks. Overall I like this paper and believe it should be accepted. Strengths: 1. The paper is written fairly clearly and is easy enough to understand. 2. It is a nice idea, exploiting tree structure in the ground metric, and yields an efficient algorithm. 3. The empirical results in Section 6 demonstrate the utility of tree-slicing. Weaknesses: 1. It appears in Sections 6.1 and 6.2 that the tree-sliced Wasserstein distance outperforms the original optimal transport distance, which is surprising. Could you explain why this occurs? 2. The proof in the main text of Proposition 1 looks more like a proof sketch, particularly as the existence of a function f having the property you claim isn't immediately obvious. Could you include (in the supplement, at least) the full proof? --- UPDATE: I have read and I appreciate the authors' response. I will not be changing my score.

Reviewer 2



The paper introduces the three-slice Wasserstein metric as a generalization of the well known sliced-Wasserstein metric. The new metric can be computed very efficiently and it is more flexible than the sliced metric. The underlying idea is simple and appealing and it can potentially have a sizable impact on the generative modeling and variational inference. The paper also introduces a new positive-definite kernel. In general, I think that the contributions of the paper are relevant and original. The writing is clear and the basic ideas are properly presented to the reader. However, I have some concerns concerning the theoretical results and the experimental analysis. Major comments: - Most of the theoretical results (proposition 1 and 2) are not original as they follow directly from previous work on the tree OT metrics. In my opinion, the bound on the Wasserstein distance given at line 185 is an interesting result as it partially justifies the minimization of the sliced-tree metric as a proxy for the minimization of the intractable Wasserstein metric. However, this result should be complemented with further analysis concerning the tightness of the bound and the limiting behavior for the number of trees tending to infinity under several possible tree distributions. Specifically, the properties of this limit under the hypercube sampling model deserves further investigation. In absence of theoretical result, it would be useful to run a numeric analysis comparing the limit of the hypercube tree-sliced metric to the Wasserstein metric and the sliced Wasserstein metric. - The experimental section is fairly well done. However, the first experiment on topological data analysis is a bit to esoteric and it is not easy to interpret for the many readers not familiar with the field. All experiments concern with the tree-sliced kernel but most of the paper is concerned with the metric itself. It would have been very useful to have an experiment where the metric is used in a more conventional OT problem such as color transfer or generative modeling. Minor comments: - The results of the topological analysis are scattered across too many figures. I would suggest to organize them on a single multi paneled figures. - The word embedding experiment should probably be presented before the TDA experiment as it is easier to understand for a wider audience.

Reviewer 3



The paper suggests to replace the Wasserstein distance in applications with a "tree-sliced" variant, which embeds the points into a tree metric and computes Wasserstein on the tree, which takes a closed-form solution on tree metrics and can be easily computed. In particular, the Wasserstein distance on the tree is an l_1-distance between the pointsets, and then the paper suggests to use the associated Laplacian kernel, i.e., exp(-l_1-distance). The application to Wasserstein distance is natural and also generalizes the "sliced approximation" of Wasserstein which uses 1-dimensional embeddings of the pointsets (i.e., path metrics instead of tree metrics). It should be noted however that in a broader context, embedding the input pointset in a tree metric (or average of trees) and solving the problem on the tree (where many problems become dramatically more tractable) is a standard approach that and has been applied to countless metric problems, with great success. This is not to say that instantiating it for Wasserstein distance is less significant, but to put originality in context. What is most troubling is that the paper seems to be completely unaware any literature of embedding points into a distribution over tree metrics, and claims some standard and well-known techniques and novelties. This has been a vast and prominent field of research for over two decades with countless papers; some classic references are [1,2,3], but there are many more. In particular, "algorithm 1" is an extremely standard and well-known construction called a quadtree. It is also well-known that shifting the initial hypercube at random (equivalent to your random expanding of it) preserves the distances in expectation with bounded distortion, i.e., that averaging over several such trees is a provably good probabilistic embedding (eg. [4, section 2.3]). I don't see why such a classic notion needs to be laid out in full in the main text and presented as a contribution; I can only assume this is a case of unaware "rediscovery". Indeed, what the paper calls "partition-based methods" form the bulk of literature for bounded-dimensional pointsets, and "clustering-based methods" form the bulk for general finite metric spaces [3] and in particular high-dimensional ones [2]. This part of the paper appears to warrant substantial revision. Moving on to a different topic, I found some parts of the paper overly obscure and unclear. In particular: 1. Definite-negativity is mentioned and highlighted so many times, that the notion should probably be defined in the paper and not just by reference, and also perhaps explain why is it important to you. Is this to ensure that the kernel is positive-definite? 2. I am unclear on where slicing comes into play. The TW kernel is defined for one tree, and afterwards you define STW citing empirical motivation. So, which kernel did you actually use in the experiments -- exp(-TW) or exp(-STW)? If it's the former, then why define STW, and what are the empirical considerations mentioned in line 107? If it's the latter, then why is it apparently not mentioned (and the kernel is called k_TW and not k_STW)? Is there a reason to use one and not the other? (I'm noting that STW is also negative-definite, simply because the average of l_1-metrics is an l_1-metric -- right?) 3. The description of the clustering-based tree construction (line 141) is too obscure. How do you set the number of clusters in each level? What is the connection to fast Gauss transform (why is it more related than any other clustering method)? Why do you use further-pair clustering (a.k.a k-center) instead of, say, the more standard k-means, or any other method? Does it have downstream motivation and does it effect the results? Conclusion: The upside is that idea at the base of this paper, while simple, is nice and potentially useful, and the experiments show advantage. I do like the suggested approach and find it interesting. On the downside, the paper has a significant issue with wheel-reinventing and apparent unfamiliarity with relevant literature, and some overly obscure parts in the presentation. ==Update== I have read the authors' response. It was somewhat difficult to understand and I am unsure of the extent to which it acknowledges the issue of duplication of prior work. I emphasize that there are prior works on embedding specifically OT into tree metric: For example Indyk-Thaper [5], ref [37] in your submission, with the same tree-of-hypercubes construction, l_1-embedding of the resulting tree metric, and application to nearest neighbor search. See in particular their "our techniques" section and more references therein (eg. Charikar'02). I would strongly advise to revise the paper so as to correctly position it in the existing literature with due attribution, and highlight its actual contributions which are--as I see it--exploring the empirical application of these techniques to real-world classification tasks (whereas the aforementioned prior work was more theoretically-minded), and in particular using the exponential kernel associated with the l_1-embedding of the tree metric. References: [1] Y. Bartal, Probabilistic approximation of metric spaces and its algorithmic applications, FOCS 1996. [2] M. Charikar, C. Chekuri, A. Goel, S. Guha, S. Plotkin, Approximating a finite metric by a small number of tree metrics, FOCS 1998. [3] J. Fakcharoenphol, S. Rao, K. Talwar, A tight bound on approximating arbitrary metrics by tree metrics, JCSS 2004. [4] P. Indyk, Algorithmic applications of low-distortion geometric embeddings, FOCS 2001. [5] P. Indyk, N. Thaper, Fast image retrieval via embeddings, SCTV 2001.

[Author Response · NeurIPS 2019]

Reviewer #1 (R1): ♦*[...]tree-sliced Wasserstein distance outperforms the original optimal transport distance[...]* We used the tree-sliced Wasserstein (TSW) and OT distances within an RBF kernel, $k = \exp(-td)$. The TSW kernel is p.d. while the OT kernel $k_{\mathrm{OT}}$ is not. The indefiniteness of $k_{\mathrm{OT}}$ may affect its performances in some applications: $k_{\mathrm{OT}}$ performs worse in TDA applications in §6.1, but works well for documents with word embedding applications in §6.2. ♦*[...]include (in the supplement, at least) the full proof [of Proposition 1]?* Agreed, we will follow your suggestion.

Reviewer #2: ♦*[...]Prop. 1 and 2 are not original[...]* ♦R1: *proof of negative definiteness of the proposed OT[...]* We do state that the proof of Prop. 1 is not original in $\ell$.39–41. Although Prop. 2 follows from Prop. 1, it follows the idea underlying sliced W kernels (or Gaussian processes as you mention), and this result remains new to our knowledge.

♦*[...]bound on the Wasserstein distance[...]interesting result[...]numeric analysis comparing the limit of the hypercube tree-sliced metric to the Wasserstein metric and the sliced Wasserstein metric.*♦R1: *An upper bound on the Euclidean OT[...]* The hypercube tree-sliced metric is our suggestion to build practical tree metrics for TSW when used on low-dimensional data spaces (e.g. TDA in §6.1). We insist that we do not try to mimic the Euclidean OT ($W_2$) or the sliced-Wasserstein (SW), but rather propose a variant of OT distance. As stated in $\ell$.173–179, and $\ell$.158–161 in §5, SW is a special case of TSW. Following your point, we have carried out the following experiment: for a query point $q$, let $p$ be its nearest neighbor w.r.t. TSW. Figure 1 illustrates that $p$ is very likely among the top 5 (on MPEG7), and top 10 (on Orbit) near neighbors on the $W_2$ space (results are averaged over 1000 runs of random split $90\%/10\%$ for training and test). When the number of tree-slices in TSW increases, the $W_2$ near-neighbor rank of $p$ is improved. These empirical results suggest that TSW may agree with some aspects of $W_2$.

Figure 1: A frequency of near-neighbor rank on the $W_2$ space for the nearest neighbor w.r.t. TSW.

♦*[...]experiment where the metric is used in a more conventional OT problem such as color transfer or generative modeling.* In the experiments, we used RBF kernels ($k = \exp(-td)$ for a given metric $d$) with SVM which usually improves on $k$-NN results. We will add $k$-NN results. We are now considering color transfer and barycenter applications. Gradients of TSW w.r.t. supports and weights of empirical measures can be recovered pending some choices in how interpolations are defined.♦*a single multi paneled figure [...] word embedding experiment[...]presented before the TDA* Many thanks for your suggestions. We will incorporate them.

Reviewer #3: ♦*What is most troubling is that the paper seems to be completely unaware any literature of embedding points into a distribution over metrics, and claims some standard and well-known techniques and novelties[...]* We understand your point and will do everything to correct this misunderstanding. This was caused by a lack of care in the presentation of §4. This was not the message we wanted to convey. We will rewrite this section following your comments. As you have gathered from our algorithms, approximating an arbitrary metric using trees is *not* a key goal in our submission, our goal is stated in $\ell$.41–44 in §1. Much like 1D projections do not offer interesting properties from a distortion perspective but remain useful for SW, we do believe that trees with large distortion can still remain useful. This is because metric approximations are used within another computation (Wasserstein) and therefore we do not gain from overfitting too much our trees so that they match the true metric, as long as they provide guidance on the optimal assignment. We will insist more on the importance of sampling tree metrics randomly, both for low-dimensional in §6.1 *and* high-dimensional §6.2 regimes. ♦*Definite-negativity is mentioned and highlighted[...] explain why is it important to you. Is this to ensure that the kernel is positive-definite?* Negative definiteness of a distance means essentially that the space is flat and that positive definite kernels can be easily derived from them, following Berg et al.'s theorem. This is why kernel methods kick in from §.6 (or Gaussian processes as per Reviewer #2's suggestion). We will clarify this motivation following your comment. ♦*[...]which kernel did you actually use in the experiments – exp(-TW) or exp(-TSW)?[...]kernel is called $k_{\mathrm{TW}}$ and not $k_{\mathrm{TSW}}$?[...]TSW is also negative-definite, simply because the average of $l_1$-metrics is an $l_1$-metric – right?)* In the experiments, we used the kernel $\exp(-td_{\mathrm{TSW}})$ as stated in $\ell$.225–227 for §6.1, and $\ell$.289–292 for §6.2. We will define $k_{\mathrm{TSW}}$, and rename $k_{\mathrm{TW}}$ to $k_{\mathrm{TSW}}$ in the experiments following your suggestion. Indeed, averaging of negative definite functions is trivially negative definite. Hence, $d_{\mathrm{TSW}}$ is negative definite. We will clarify it in the updated version. ♦*[...]set the number of clusters in each level?[...]connection to fast Gauss transform [...]more related than any other clustering method[...]use farthest-point clustering[...]downstream motivation and does it effect the results?* We fixed the same number of clusters $\kappa$ when performing the farthest-point clustering (replace step 9 in Algorithm 1) at different height levels. $\kappa$ is typically chosen via cross-validation. Moreover, we also illustrate the effect of $\kappa$ in applications in Figure 5 (and Figures 5–7 in the supplement). In general, one can apply any favorite clustering methods. We used the farthest-point clustering due to its fast computation, i.e. $O(n \log \kappa)$ as stated in $\ell$.148–149 to construct practical tree metrics for applications with high-dimensional data space, e.g. documents with word embedding applications in §6.2.

[Meta-Review · NeurIPS 2019]

The idea of embedding the Wasserstein distance in a tree-metric is an interesting idea accompanied with a thorough study and showing good empirical results. The contribution could make the Wasserstein community more aware of tree metric techniques. However, the presentation of the state of the art and the position of the contribution must updated in a correct way and we urge the authors to do it in an appropriate manner. In particular, the fact that (i) any metric can be embedded into a random ensemble of trees with small distortion, (ii) a good way to do it tree-of-hypercubes, (iii) this is useful for approximating OT and for nearest neighbor search in the OT metric, (iv) OT on trees is an l_1-metric are clearly contributions related to ref [37]. The appropriate reference for algorithm 1 must also be clearly mentioned. Authors should take into account the remarks and suggestions made by reviewer 3 and check appropriately the literature. The position of the contribution with respect to the existing literature must be highlighted clearly. This is beyond a classic camera-ready copy work, but this is essential from a scientific point of view. Many thanks for your consideration.